# Ketone Bodies Are Mildly Elevated in Subjects with Type 2 Diabetes Mellitus and Are Inversely Associated with Insulin Resistance as Measured by the Lipoprotein Insulin Resistance Index

**DOI:** 10.3390/jcm9020321

**Published:** 2020-01-23

**Authors:** Erwin Garcia, Irina Shalaurova, Steven P. Matyus, David N. Oskardmay, James D. Otvos, Robin P.F. Dullaart, Margery A. Connelly

**Affiliations:** 1Laboratory Corporation of America Holdings (LabCorp), Morrisville, NC 27560, USA; shalaui@labcorp.com (I.S.); matyuss@labcorp.com (S.P.M.); oskardd@labcorp.com (D.N.O.); otvosj@labcorp.com (J.D.O.); connem5@labcorp.com (M.A.C.); 2Department of Endocrinology, University of Groningen and University Medical Center Groningen, 9700 RB Groningen, The Netherlands; r.p.f.dullaart@umcg.nl

**Keywords:** ketone bodies, nuclear magnetic resonance spectroscopy, type 2 diabetes mellitus, insulin resistance

## Abstract

Background: Quantifying mildly elevated ketone bodies is clinically and pathophysiologically relevant, especially in the context of disease states as well as for monitoring of various diets and exercise regimens. As an alternative assay for measuring ketone bodies in the clinical laboratory, a nuclear magnetic resonance (NMR) spectroscopy-based test was developed for quantification of β-hydroxybutyrate (β-HB), acetoacetate (AcAc) and acetone. Methods: The ketone body assay was evaluated for precision, linearity and stability and method comparisons were performed. In addition, plasma ketone bodies were measured in the Insulin Resistance Atherosclerosis Study (IRAS, *n* = 1198; 373 type 2 diabetes mellitus (T2DM) subjects). Results: β-HB and AcAc quantified using NMR and mass spectrometry and acetone quantified using NMR and gas chromatography/mass spectrometry were highly correlated (R^2^ = 0.996, 0.994, and 0.994 for β-HB, AcAc, acetone, respectively). Coefficients of variation (%CVs) for intra- and inter-assay precision ranged from 1.3% to 9.3%, 3.1% to 7.7%, and 3.8% to 9.1%, for β-HB, AcAc and acetone, respectively. In the IRAS, ketone bodies were elevated in subjects with T2DM versus non-diabetic individuals (*p* = 0.011 to ≤0.001). Age- and sex-adjusted multivariable linear regression analysis revealed that total ketone bodies and β-HB were associated directly with free fatty acids (FFAs) and T2DM and inversely with triglycerides and insulin resistance as measured by the Lipoprotein Insulin Resistance Index. Conclusions: Concentrations of the three main ketone bodies can be determined by NMR with good clinical performance, are elevated in T2DM and are inversely associated with triglycerides and insulin resistance.

## 1. Introduction

Ketone bodies are products of fat catabolism that are used as alternative substrates to glucose as sources of energy when carbohydrate intake is low and there is a surplus of circulating free fatty acids (FFAs) [1]. Elevations in ketone bodies occur when there is insufficient insulin to assist in the utilization of glucose as an energy source (ketoacidosis), during times of prolonged fasting (mild ketonemia) or when consuming a carbohydrate restricted diet (ketogenic diet). The predominant ketone bodies are β-hydroxybutyrate (3-hydroxybutyrate; β-HB), acetoacetate (AcAc) and acetone. Under normal physiological conditions, total plasma ketone concentrations fluctuate between 100 and 600 µM but can rise to ~1 mM after prolonged exercise or 24 h fasting. Diabetic ketoacidosis is a serious acute complication that occurs in patients with diabetes and is diagnosed by having ketone body concentrations >1–1.5 mM. A less common cause of ketoacidosis is due to alcohol abuse, which usually presents in malnourished patients with a history of a recent alcoholic binge [2,3]. Total ketone body concentrations can rise as high as 6–20 mM in patients with diabetic or alcoholic ketoacidosis [4,5]. 

While measurement of ketone body concentrations is critical for diagnosis of ketoacidosis, and rapid (bedside) tests are widely available, it may also be useful to quantify circulating ketone bodies when there are only mild elevations. For example, diets that are rich in proteins and fats but low in carbohydrates (e.g., Atkins diet, also known as a ketogenic diet) have been shown to induce mild elevations of ketone bodies. Elevations in ketone bodies produced while consuming a ketogenic diet are much lower than those observed in ketoacidosis and are not associated with a change in blood pH [6]. Furthermore, ketone bodies may add value to biomarker panels for discerning the metabolic pathways that are altered in individual patients on exercise, diet or lifestyle management plans as well as on drug therapy [6,7]. In particular, sodium–glucose cotransporter-2 (SGLT2) inhibitor treatment for T2DM may elicit ketonemia that is reminiscent of the mild elevations induced by ketogenic diets. These mild elevations may be related to the ability of SGLT2 inhibitors to reduce plasma glucose via renal elimination, stimulate lipolysis, elicit weight loss and enhance insulin sensitivity [8,9,10]. Consequently, physicians or patients may want to monitor ketone bodies to ensure these diets are managed correctly and are having beneficial effects while not causing very high ketone body levels [11]. Such assays may also be clinically useful while patients are on SGLT2 inhibitor therapy.

Assessment of ketone body concentrations for the diagnosis of ketoacidosis is usually accomplished using point-of-care urine or blood test strips. These test strips will indicate whether your ketone body levels are in the normal range or whether they are elevated. However, they are not sensitive enough to detect mild elevations in ketone bodies that might occur while a patient is on a ketogenic diet or taking medications that raise ketone body concentrations. In the clinical laboratory, a colorimetric-spectrophotometric assay can be used as a standalone assay to measure ketone bodies. A research nuclear magnetic resonance spectroscopy (NMR) platform has also been used for quantification of ketone bodies and clinical associations were observed [12]. The potential benefit of using an NMR-based assay in the clinical laboratory is the ability to simultaneously quantify ketone bodies, glucose, amino acids, lipoproteins and other metabolites, all of which are altered with diet and drug therapy, from a single NMR spectrum acquired from a serum or plasma specimen. The NMR signals from the ketone bodies arise from the methyl group protons, and the positions of these signals within the NMR spectrum are unique and have been well characterized. In addition, the amplitudes of the signals are proportional to their respective concentrations, which makes quantification fairly straightforward.

The aims of the current study were to: (1) develop an assay for quantification of the three main ketone bodies, β-hydroxybutyrate (β-HB), acetoacetate (AcAc) and acetone on a clinical NMR instrument, (2) to evaluate its analytical performance, and (3) to assess the clinical associations of ketone bodies with dysglycemia, and insulin resistance in a large, multi-ethnic population.

## 2. Methods

### 2.1. Materials and Specimen Collection

β-HB, AcAc and acetone were purchased from Sigma Aldrich (St. Louis, MO, USA). Serum pools were prepared by identifying and pooling both fasting and non-fasting serum samples with high and low ranges of the three ketone bodies from donor subjects (LabCorp, Morrisville, NC, USA) or by spiking. 

### 2.2. The Insulin Resistance Atherosclerosis Study (IRAS)

The Insulin Resistance Atherosclerosis Study (IRAS) recruited 1625 participants from four clinical centers located in San Antonio, TX; San Luis Valley, CO; Oakland, CA; and Los Angeles, CA, between October 1992 and April 1994. Details of the study population, research methods and exclusion criteria have been published previously [13]. T2DM was defined as fasting plasma glucose (FPG) concentration ≥7.0 mmol/L and/or 2 h glucose concentration ≥11.1 mmol/L by a 75 g oral glucose tolerance test (OGTT) using World Health Organization criteria [14]. Demographic (e.g., age, sex, ethnicity) and lifestyle factors (e.g., smoking, alcohol consumption) were collected on standardized questionnaires by self-report [13,15]. Details of the relevant clinical procedures and laboratory measurements were described previously [16]. NMR spectra were acquired in the year 2000 from EDTA plasma samples that were collected after an overnight fast at baseline (1992–1994) (LabCorp, Morrisville, NC, USA) [14,17]. Samples were stored frozen at <−70 °C until the time of testing (6–8 years). An internal stability study revealed that the ketone bodies were stable when frozen at <−70 °C for 12 years (<15% bias) which is longer than the time the IRAS samples were frozen before the NMR spectra were acquired. Quantification of the three ketone bodies was accomplished by reanalyzing the digitally stored NMR spectra using the newly developed ketone body assay software algorithm. The sample size of the current report was 1198 participants after exclusion of subjects that were missing NMR or covariate data. All participants provided written informed consent and the study was performed according to the principles outlined in the Declaration of Helsinki. The institutional review boards (IRB) at each study site approved the study protocol. 

### 2.3. Acquisition of NMR Spectra

The NMR spectra used to quantify the ketone bodies were collected on a Vantera^®^ Clinical Analyzer (LabCorp, Morrisville, NC, USA), a fully automated, high-throughput, (^1^H) NMR platform, in the same manner as the NMR LipoProfile test and branched chain amino acids [16,18,19,20]. The Vantera is equipped with a 400 MHz (9.4 T) Agilent spectrometer, a 4 mm indirect detection probe and a flow cell that was equilibrated at 47 °C. The 1D ^1^H NMR spectra were collected. The water resonance was attenuated using the WET solvent suppression technique as described previously [19,21]. Each NMR spectrum was acquired for a total of 48 s (9024 data points, 4496.4 Hz spectral width, 2.95 s relaxation delay between scans, 12 scans). The free induction decay signal was zero-filled to 32,768 points and multiplied by a Gaussian function (for resolution enhancement) prior to Fourier Transformation.

### 2.4. Quantification of Ketone Bodies by Signal Deconvolution Analysis

The methyl signals from the three ketone bodies produce distinct peaks in the ^1^H NMR spectrum that can be used for quantification. However, these signals overlap with the NMR signals that arise from circulating proteins and lipids in the serum sample (Figure 1). Spiking experiments with pure β-HB, AcAc and acetone in serum confirmed the positions of their signature peaks (Figure 1, insets A and B). For quantification of the NMR signals for the ketone bodies, a new assay was developed that mathematically models the peaks for the ketone bodies as well as the underlying background signals from the lipoproteins, proteins and small molecules in two spectral regions (between 1.14 and 1.17 ppm for β-HB and 2.18 and 2.24 ppm for AcAc and acetone) and reports levels of β-HB, AcAc, acetone and total ketone bodies. β-HB can be quantified using signals at 1.13–1.19 ppm as well at 2.33 and 4.13 ppm. The β-HB signal at 1.13–1.19 ppm was used for quantification because of its simplicity (doublet) and higher intensity (corresponding to 3 protons) compared to the signals at 2.33 ppm (signal split into 8 peak multiplet components; corresponding to 2 protons) or 4.13 ppm (signal split into 6 peak multiplet components; corresponding to 1 proton). In addition, in clinical samples it was observed that there were more peaks (some unknown) overlapping with the β-HB resonance at 2.33 ppm which made it difficult to quantify on its own. Furthermore, variability in water signal suppression, in addition to peak overlap, could affect deconvolution when using the β-HB resonance at 4.13 ppm. Moreover, using the signal at 1.13–1.19 ppm had the advantage of using a spectral region to measure β-HB that was separate from the region used to quantify acetoacetate and acetone. The spectral region of interest in the frequency domain was deconvoluted into its parts by using a non-negative linear least squares algorithm. The resulting spectral model was generated with the composite lineshapes for the signals corresponding to ketone bodies and the addition of components comprising each of the background signals such that the final model coincides closely to the observed NMR signal. As such, the signals corresponding to the background are, in essence, subtracted during deconvolution and the signal areas for ketone bodies are subsequently multiplied by their conversion factors to yield concentration units. Since the peak area correlates linearly with concentration (Figure 2), the conversion factor for each analyte was determined by calculating the analyte signal area from spectra obtained for dialyzed serum that were spiked with a known amount of each standard, and relating the area to the expected concentration of the analyte. The units for the 3 ketone bodies are µM. 

### 2.5. Determination of LP-IR Scores and GlycA Levels

The Lipoprotein Insulin Resistance Index (LP-IR) is an NMR-based assay that produces scores that range from 0 to 100 (from least to most insulin resistant) [22]. LP-IR scores are calculated using six lipoprotein parameters that are quantified by NMR: very-low-density lipoprotein (VLDL), low-density lipoprotein (LDL) and high-density lipoprotein (HDL) size, along with concentrations of large VLDL, small LDL and large HDL particles [22]. In addition to LP-IR results being correlated with other indices of insulin resistance such as Homeostatic Model Assessment of Insulin Resistance (HOMA-IR) and the sensitivity index (Si), LP-IR scores have been shown to predict future T2DM in multiple study populations [23,24,25,26]. GlycA is an NMR-based marker of systemic inflammation whose signal arises from highly glycosylated acute phase proteins (e.g., α1-acid glycoprotein, haptoglobin, α1-antitrypsin, and α1-antichymotrypsin) [27]. GlycA levels were quantified as previously described and have been shown to be related to T2DM and cardiovascular disease [27,28,29]. LP-IR scores and GlycA levels were derived from the same NMR spectra from which ketone body concentrations were determined.

### 2.6. Assay Performance Testing

Dialyzed (through a Slide-A-Lyzer 10 kDa molecular weight cutoff cassette (Thermo Scientific, Rockford, IL, USA)) serum proteins isolated by ultracentrifugation from five pooled, de-identified residual serum specimens were used for determining the limits of blank (LOB). Five serum pools containing low concentrations of β-HB, AcAc and acetone were tested to determine the limits of detection (LOD) and 12 or more serum pools (AcAc: 18 pools; acetone: 15; β-HB: 12; total ketone bodies: 16) were used to determine the limit of quantitation (LOQ) according to Clinical and Laboratory Standards Institute (CLSI) guidelines [30], as previously described [19]. The LOQ was determined at 20% CV for β-HB and 30% CV for AcAc and acetone, the latter due to the poor precision at low concentrations and instability. Consistent with CLSI guidelines [31], linearity was evaluated by identifying serum pools with low and high levels of the ketone bodies and mixing them with the intent of covering the known biological ranges. In addition, serum pools were spiked individually with stocks of β-HB, AcAc and acetone in order to achieve the higher concentrations. Mean concentrations and coefficients of variation (%CVs) were calculated for each pool. Within-run and within-laboratory imprecision were determined based on CLSI guidelines [32]. Low, intermediate and high pools were tested for β-HB in order to include a level near the 1–1.5 mM clinical decision point used for diagnosing ketoacidosis and low and high pools were used for AcAc and acetone. Within-run (intra-assay or within run) imprecision was determined by analyzing each of the pools on one day and one instrument (*n* = 20). The same pools were analyzed for 20 days, two replicates twice per day, (total *n* = 80) to evaluate the within-laboratory (inter-assay or between runs) imprecision. In order to avoid instability issues, especially for AcAc and acetone, aliquots of the various pools were kept frozen until testing.

### 2.7. Method Comparison

Method comparison studies, consistent with CLSI guidelines [33], were performed to compare ketone body quantification by NMR versus liquid chromatography coupled to tandem mass spectrometry (LC/MS/MS) or gas chromatography/mass spectrometry (GC/MS). The LC/MS/MS analysis was performed on a Waters Quattro Premier XE employing the High Performance Liquid Chromatography tandem Mass Spectrometry (HPLC-tMS) method in the Multiple Reaction Monitoring (MRM) setting. The internal standard solution was prepared in 0.1% formic acid. Each sample (100 µL) was spiked with 50 µL internal standard solution, vortexed (5 min) and incubated at 4 °C for 10 min. After incubation, 250 µL of 70% HClO_4_ was added to each sample, vortexed (10 min) and incubated at 4 °C for 10 min. The samples were then centrifuged at 13,400× *g* at 4 °C for 10 min. A 400 µL clear supernatant was transferred into the HPLC autosampler vial for LC/MS/MS analysis. For the GC/MS analysis, a Shimadzu 2010 plus GC system was used with a QP-2010 mass detector and an AOC5000 auto-sampler. The MS parameters were: *m*/*z* +43 for quantify and +58 for reference. The ion source (EI) temperature was 230 °C and the GC parameters included a DB-WAX 30 m × 250 µm × 0.25 µm column with a flow rate of 1.0 mL/min and a column temperature of 160 °C. Standard curves were run at the beginning and end of each run and quality control checks were run every 8 samples. EDTA plasma specimens were obtained from 50 donors and aliquots were immediately frozen at <−70 °C until the time of analysis. Some samples were spiked with the three analytes in order to reach the highest concentrations. The same frozen plasma samples were analyzed via NMR (LabCorp, Morrisville, NC) and LC/MS/MS (Creative Proteomics, Shirley, NY, USA) for β-HB and AcAc (*n* = 50) or GC/MS for acetone (*n* = 27) (Creative Proteomics, Shirley, NY, USA). Deming regression analysis and Bland–Altman plots were used to evaluate the correlations between the results from the different platforms.

### 2.8. Comparison of Specimen Collection Tubes and Stability Testing

Blood from 27 donors was drawn into four different tubes: a black and yellow-top Greiner serum collection tube (part #456293P), also known as a LipoTube (Greiner Bio-One, Monroe, NC, USA), as well as a red-top BD Vacutainer plain serum tube (no gel barrier), a purple-top K_2_EDTA plasma tube and a green-top sodium heparin tube (Becton Dickinson, Franklin Lakes, NJ, USA). In order to expand the range of measured values, 10 specimens were spiked with stocks of the 3 ketone bodies (<5% by volume) for a total of 37 specimens. Results from samples collected in plain serum, EDTA plasma and heparin plasma tubes were compared to results from the Greiner LipoTubes by linear regression. Specimens from 9 donors (3 donors per analyte; drawn in LipoTubes) were used to assess specimen/analyte stability. Serum samples were stored at room temperature or refrigerated (4–8 °C) and aliquots were tested hourly and daily. Specimens were also tested after freezing at <−70 °C for 15 days and after 3 freeze–thaw cycles. Mean results for all donors were evaluated with acceptable differences falling within ±10% of the day 0 (draw day) mean. To assess long-term frozen stability, EDTA plasma samples from an apparently healthy study population were tested and then frozen at <−70 °C for 3 and 6 years. 

### 2.9. Reference Interval Studies

To determine the reference intervals for the analytes reported by the ketone body assay, samples were analyzed from a cohort of generally healthy adult men and non-pregnant women between the ages of 18 and 84 who were recruited as volunteer donors. The non-fasting serum samples used for this analysis were collected in Greiner LipoTubes (described above) and processed as prescribed by the manufacturer. A description of this study population has been reported [19]; subjects that were excluded from the original study based on body mass index (BMI) and blood pressure criteria or whether or not they were taking lipid altering medications were not excluded from this study. The number of subjects used for this analysis was 552; 337 females and 215 males. The NMR spectra for these study samples were acquired on fresh samples and then stored digitally for potential future use. Quantification of the three ketone bodies was accomplished by reanalyzing the digitally stored NMR spectra using the newly developed ketone body assay software algorithm. The reference intervals were estimated using non-parametric analyses with reference limits at the 2.5th and 97.5th percentiles according to CLSI guidelines [34]. Reference intervals were compared by assessing their median results using the Wilcoxon–Mann–Whitney and Kruskal–Wallis tests. Reference intervals were confirmed in 177,000 NMR spectra collected over several months in the clinical NMR laboratory from fasting and non-fasting individuals whose serum samples were received for the NMR LipoProfile test. The protocol for collection of the samples for this study was approved by an IRB.

### 2.10. Statistical Analyses

Statistical analyses were performed using SAS v9.4 (SAS Institute, Cary, NC, USA) and Analyze-it v5.10 (Analyze-it Software, Ltd., Leeds, UK). For the analytical validation studies, linear regression analyses were performed for comparisons between variables. For the epidemiological studies, data are expressed as the mean ± SD (or SEM for figures) or as the median (interquartile range). Skewed variables were natural log (Ln) transformed. Between-group differences in continuous variables were determined by unpaired *t*-tests. Between-group differences in dichotomous variables were determined by χ^2^-analysis. Univariate analyses were conducted using Spearman correlation coefficients. Multivariable linear regression analyses were performed to determine the independent associations of total ketone bodies, β-HB, AcAc and acetone with various clinical parameters. Two-sided *p*-values < 0.05 were considered statistically significant. 

## 3. Results

### 3.1. Ketone Body Assay Development and Performance Characteristics

The signals from the methyl groups of β-HB, AcAc and acetone overlap with signals from proteins, carbohydrate residues and lipoproteins in the 1D ^1^H NMR spectrum of serum (Figure 2). Therefore, a new NMR-based deconvolution assay was developed that mathematically quantifies the signals from, and concentrations of, each individual ketone body in the context of the lipoprotein and protein signals in each plasma or serum sample (Figure 2). The limits of blank (LOB) for total ketone bodies, β-HB, AcAc and acetone as reported by this assay were determined to be 40.1, 27.9, 15.7 and 5.6 µM, respectively. The analytical sensitivity or limits of detection (LOD) were calculated to be 61.2, 45.0, 24.8 and 11.2 µM. The functional sensitivity or limit of quantitation (LOQ) for total ketone bodies, β-HB, AcAc and acetone was determined to be 65.0, 45.0, 26.3 and 19.7 µM, respectively. Linearity was demonstrated between 51.0 and 10,445 µM for β-HB, 52.0 and 4715 µM for AcAc and 6.0 and 8943 µM for acetone, which span the range of values typically observed in the clinical laboratory (Appendix A). The correlation coefficients (*R*^2^) were 1.00 for all three analytes. Based on the linearity and LOQ data, the reportable ranges for β-HB, AcAc and acetone are 51.0 and 10,445 µM (0.53–108.7 mg/dL), 52.0 and 4,715 µM (0.53–48.1 mg/dL) and 19.7 and 8,943 µM (0.03–51.9 mg/dL), respectively. Sample pools with low and high levels for AcAc and acetone or low, intermediate and high levels for β-HB were tested for intra- (within-run) and inter-assay (within-lab) precision. For β-HB, AcAc and acetone, the CVs for intra-assay and inter-assay precision were 1.3%–9.3%, 3.1%–7.7%, and 3.8%–9.1%, respectively (Table 1). 

A method comparison study was performed comparing quantification by NMR to platforms commonly used for determining ketone body concentrations—LC/MS/MS for β-HB and AcAc and GC/MS for acetone. A comparison of plasma concentrations using the comparator platforms correlated well by Deming regression with R^2^ values of 0.996, 0.994 and 0.994 for β-HB, AcAc and acetone, respectively (Figure 3A–C). The Bland–Altman plots revealed that the residuals, while randomly dispersed, showed a slight bias toward higher results for β-HB and acetoacetate when quantified by NMR compared to the LC/MS/MS assay (Figure 3D–E). For acetone, the residuals were not randomly dispersed, and they trended downward at the higher concentrations. In addition, there was a bias toward lower results when quantified by NMR compared to the GC/MS assay (Figure 3F). However, when standard curves were used to evaluate the accuracy of the different methods, the NMR results more closely reflected the known concentrations of the three ketone bodies (data not shown). For all three analytes, only one specimen can be considered outside the limits of agreements (LOAs).

Four types of specimen collection tubes were evaluated for their suitability in the ketone body assay. Results from specimens collected in plain red-top serum, EDTA plasma and heparin plasma tubes were plotted against LipoTube results and linear regression analyses were performed. Appendix A summarizes the linear regression characteristics for the tube comparisons. Results indicate that there was minimal (<2%) bias for β-HB and acetone in plain serum, EDTA plasma or heparin plasma tubes versus serum collected in LipoTubes for β-HB and acetone. AcAc levels were somewhat lower in all three tubes versus LipoTubes. 

The stability of the three ketone bodies was evaluated in serum samples stored at room temperature, refrigerated (2–8 °C), or frozen (<−70 °C). Results demonstrated that β-HB was stable up to 48 h at room temperature and up to 72 h when refrigerated (2–8 °C) (Appendix A). AcAc and acetone were stable up to 15 days when frozen <−70 °C. While acetone was stable when stored at room temperature for 8 or 24 h, or when refrigerated for 24–48, AcAc was not. All three analytes were stable for up to three freeze–thaw cycles. While AcAc and acetone were unstable (Bias = −28.2 and 22.5%, respectively, after 3 years and −13.8 and 19.1%, respectively, after 6 years), β-HB was stable for 3 and 6 years when frozen at <−70 °C (Bias = 0.0% and −1.5%).

A population of generally healthy adult men and women (*n* = 552) was used to determine the reference intervals for the analytes reported by the ketone body assay. Table 2 shows the distribution of values for total ketone bodies, β-HB, AcAc and acetone in this population. For total ketone bodies, β-HB, AcAc and acetone, the mean ± the standard deviation (SD) were 212.9 ± 133.8, 139.3 ± 91.5, 42.5 ± 28.8 and 31.2 ± 22.8 µM and the reference intervals were 88.8–623.2, 48.6–396.4, 26.3–120.0 and 19.7–78.6 µM, respectively (Table 2). There were no differences detected between the medians in the women (*n* = 337) versus the men (*n* = 215) in this population (data not shown).

To confirm the reference intervals, we reanalyzed NMR spectra (*n* = 177,000) collected over several months in the clinical NMR laboratory from fasting and non-fasting individuals whose serum samples were sent for the NMR LipoProfile test. The distributions of the concentrations for total ketone bodies, β-HB, AcAc and acetone in these subjects can be found in Appendix A. The means ± the SD in this population were 287.0 ± 365.7, 193.6 ± 252.4, 48.9 ± 46.5 and 44.5 ± 84.7 µM and the reference intervals were 87.5–987.1, 49.5–693.9, 26.3–143.4 and 19.7–179.5 µM for total ketone bodies, β-HB, AcAc and acetone, respectively (Appendix A). 

### 3.2. Cross-Sectional Analysis of Ketone Bodies in a Cohort of Subjects with Metabolic Disease

In the IRAS, T2DM subjects were older compared to non-diabetic subjects (Table 3). T2DM subjects were also more likely to have higher systolic blood pressure, BMI, waist circumference, fasting glucose, 2 h glucose, free fatty acids (FFAs), triglycerides and GlycA, an NMR-based marker of systemic inflammation that is associated with T2DM and cardiovascular disease [28,29]. They were also more likely to have lower HDL-C and impaired insulin sensitivity, the latter exemplified by having higher mean LP-IR scores. In addition, total ketone bodies, β-HB, AcAc and acetone were significantly higher in IRAS participants with T2DM compared to those who did not have T2DM at study start (Table 3). 

Univariate analysis demonstrated that in the IRAS, total ketone bodies, β-HB, AcAc and acetone were directly associated with FFAs, glucose and T2DM. Total ketone bodies and β-HB were inversely associated with TG, and insulin resistance as measured by LP-IR (Table 4). This was true in both non-diabetic and diabetic IRAS participants.

Age- and sex-adjusted multivariable linear regression analysis demonstrated that total ketone bodies, β-HB, AcAc and acetone were independently associated with FFAs and T2DM. Total ketone bodies and β-HB were inversely associated with TG and insulin resistance, as measured by LP-IR (Table 4). AcAc was directly associated with FFAs and T2DM and inversely associated with insulin resistance, but was not associated with TG. Acetone was directly associated with FFAs and T2DM and inversely associated with age, sex and insulin resistance (Table 5).

## 4. Discussion

Here, we present an NMR-based method of measuring ketone bodies by deconvolution using non-negative linear least squares algorithm. Another NMR-based method to quantify these analytes [35] has been reported. However, it differs from the assay presented here in terms of the computational method used to extract the analyte concentrations. While signal lineshape deconvolution was used for the current assay, Wang and coworkers used Bayesian regression to develop their NMR-based method. These approaches, irrespective of the computational method, can be used to measure ketone bodies by NMR as an alternative to existing test strip and colorimetric-spectrophotometric assays. 

This study is the first to evaluate a newly developed NMR-based test that quantifies ketone bodies using a deconvolution assay on a clinical NMR analyzer. It is clear from this study that if one wishes to quantify circulating AcAc and acetone, samples need to be frozen at <−70 °C immediately after collection and processing of the specimen tubes and tested immediately after thawing due to the instability of these two analytes. This is consistent with previous publications that evaluated the stability of AcAc which tends to become rapidly decarboxylated [36,37] when samples are not frozen at <−70 °C. Many protocols call for the deproteinization of samples before analysis in order to preserve AcAc. However, this is not practical for NMR analysis under clinical laboratory conditions. β-HB, on the other hand, is very stable under all conditions tested and the β-HB assay exhibited very good analytical performance characteristics. Therefore, while we are able to quantify levels of total ketone bodies, β-HB, AcAc and acetone in carefully collected and stored samples, β-HB is the preferred ketone body for clinical evaluation. One of the strengths of NMR-based tests is that after the NMR spectrum is collected, it is quick and easy to quantify multiple analytes simultaneously from the same spectrum. Therefore, while an NMR-based assay for β-HB may not be useful for point-of-care testing, it may be useful for measuring β-HB along with glucose, branched chain amino acids [16] and other diet related metabolites and nutrients [38] and presenting the results in a panel that may inform a patient or physician about the potential effects of their diet or drug treatment.

While there are guidelines for concentrations of ketone bodies and β-HB that can be used to identify individuals with diabetic or alcoholic ketoacidosis (>1.0–1.5 mM), normal reference intervals have not been well established in the literature. We sought to determine the reference intervals for total ketone bodies, β-HB, AcAc and acetone in two different populations, a generally healthy population of men and women as well as a large population of subjects whose serum samples had been sent to LabCorp’s NMR laboratory. The reference intervals in the cohort of generally healthy subjects were somewhat lower than those previously reported in 100 subjects after an overnight fast [39]. However, similar to the findings of Foster, et al., there were no significant differences between genders in the reference intervals for total ketone bodies, β-HB, AcAc or acetone [39]. While the median values for the ketones in the small and large cohorts in our study were similar, the large cohort contained a few individuals who had very high ketone levels. Notably, for uncertain reasons, 0.5%–1.0% had greater than the recommended cutoff for elevated ketone levels (>1.5 mM) with a number of patients with severe ketosis (146 subjects with total ketones >5.0 mM and 19 subjects >10.0 mM). 

To confirm that the newly developed NMR-based ketone body assay measures clinically relevant levels of ketone bodies, β-HB, AcAc and acetone were measured in the IRAS. Total ketone bodies, β-HB, AcAc and acetone were significantly higher in IRAS participants with T2DM compared to non-diabetic subjects. Additionally, total ketone bodies and β-HB were independently associated with FFAs and T2DM and inversely associated with insulin resistance. Ketone bodies are products of fatty acid catabolism that are used as alternative sources of energy to glucose when carbohydrate intake is low and there is a surplus of circulating FFAs [1]. FFAs are transported to the liver where they are oxidized to form acetyl CoA which can in part be converted to ketone bodies. Therefore, it was not surprising that total as well as the individual ketone bodies were correlated with fasting FFA levels in the IRAS. It is interesting, and possibly relevant, that higher total ketone bodies and β-HB were associated with lower insulin resistance in the IRAS cohort. The LP-IR Index was used as a measure of insulin resistance in this study. As opposed to HOMA-IR, which is dependent on glucose and insulin, LP-IR relies on NMR-measured lipoprotein parameters and is heavily weighted by large very-low-density lipoprotein (VLDL) particles [22]. The direct relationship with FFAs and the inverse relationship between total ketone bodies and TG and the LP-IR scores is not surprising in this context. In other words, as TG are catabolized as the preferred energy substrate over glucose, large VLDL will decrease as will LP-IR levels. Moreover, while the inverse relationship between ketone bodies and LP-IR was significant, the correlation was not very strong (r = −0.134, *p* < 0.0001) suggesting that there is not as close a relationship between them as there is between LP-IR and TG (r = 0.694, *p* < 0.0001). With that said, it is interesting that subjects on a ketogenic diet experience a reduction in insulin resistance and a concomitant mild increase in circulating ketone bodies [6,7]. In addition, patients on SGLT2 inhibitors to control their T2DM experience a reduction in insulin resistance, as measured by HOMA-IR, as well as a mild increase in circulating ketone bodies [8,9,10]. Taken together, these observations suggest that the mildly elevated ketone body concentrations that have been observed with consumption of a ketogenic diet or with SGLT2 treatment may reflect metabolic improvement with respect to insulin sensitivity. Furthermore, these observations may be more related to fatty acid and TG metabolism than to glucose metabolism under certain metabolic conditions.

Several strengths and limitations of the present study should be acknowledged. We consider it a strength that the assay is reliable at a low to intermediate range of ketone body values. In addition, the direct association of total ketone bodies with FFAs, and its inverse association with TG and insulin resistance was observed in a large, multi-ethnic cohort of subjects with varying degree of metabolic impairment. However, the cross-sectional design of this study does not allow one to address the nature of the observed relationships, nor to exclude the possibility of reverse causation. 

## 5. Conclusions

In conclusion, ketone body levels can be obtained by NMR with sufficient precision and accuracy for clinical use. Further, total ketone bodies were mildly elevated in patients with T2DM and were associated with fasting FFAs and inversely associated with TG and insulin resistance.

## Figures and Tables

**Figure 1 jcm-09-00321-f001:**
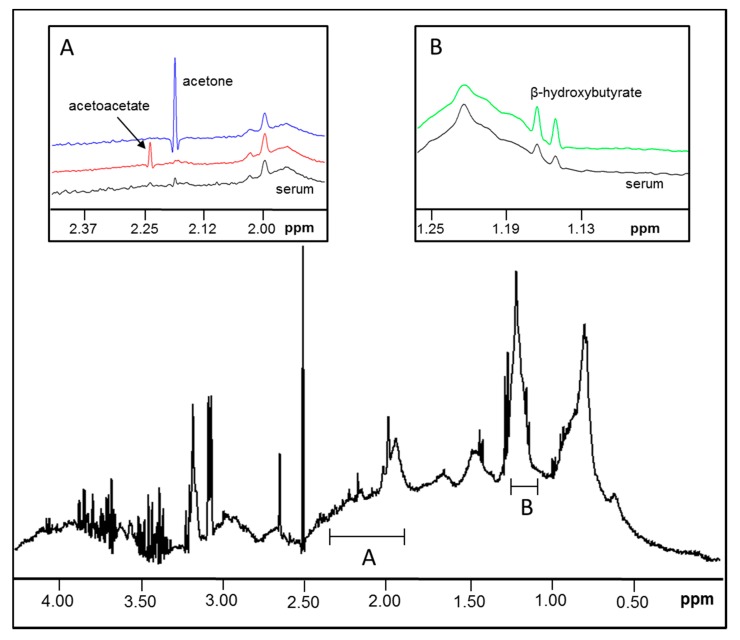
Nuclear Magnetic Resonance (NMR) signal peaks for the three ketone bodies in a spectrum of serum from a 400 MHz clinical NMR analyzer. The signals from the methyl groups on β-hydroxybutyrate (β-HB), acetoacetate (AcAc) and acetone overlap with signals that arise from the protons on lipid molecules contained within lipoprotein particles (for β-HB) and residues on the backbones and carbohydrate side-chains of proteins (for AcAc and acetone). Insets: Expanded views of NMR spectra from serum alone compared to serum spiked with either AcAc (red) and acetone (blue) (single peaks/singlet; inset (**A**) or β-HB (green; two peaks/doublet; inset (**B**) in order to confirm the relative positions and multiplicity (singlet or doublet) of the signals that were used for quantification. ppm = parts per million.

**Figure 2 jcm-09-00321-f002:**
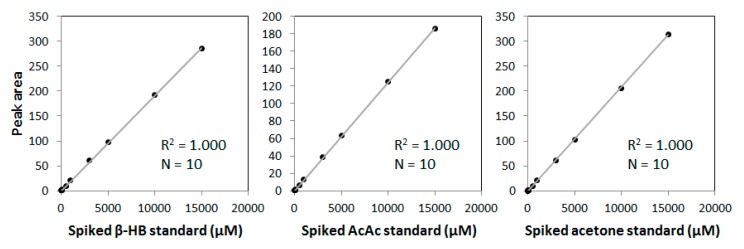
Standard curve for each ketone body used to convert peak area into µM concentrations.

**Figure 3 jcm-09-00321-f003:**
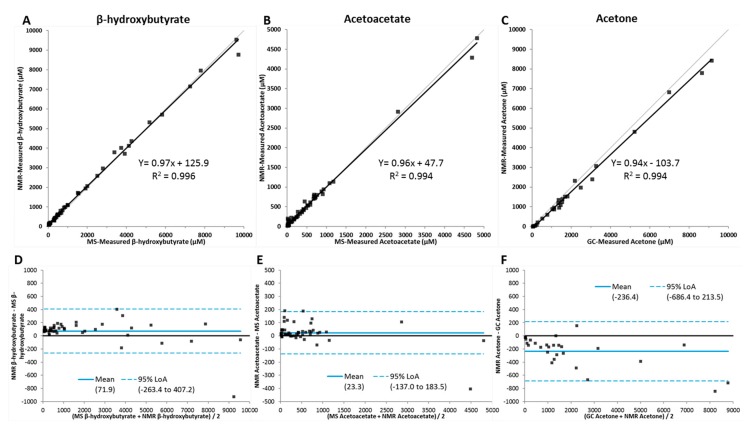
Deming regression comparison between LC/MS/MS or gas chromatography and NMR measured (**A**) β-hydroxybutyrate (*n* = 50), (**B**) acetoacetate (*n* = 50), and (**C**) acetate (*n* = 27) in serum samples. Bland–Altman plots for (**D**) β-hydroxybutyrate, (**E**) acetoacetate, and (**F**) acetone assays. The limits of agreement (LOAs) are depicted as dotted blue lines and the 0% bias is a solid black line.

**Table 1 jcm-09-00321-t001:** Within-laboratory (inter-assay) and within-run (intra-assay) imprecision for the three analytes reported by the ketone body assay (β-hydroxybutyrate, acetoacetate and acetone).

	β-hydroxybutyrate (µM)			Acetoacetate (µM)		Acetone (µM)	
	Low	Medium	High	Low	High	Low	High
**Within-lab** ^a^							
Mean	129.5	219.0	1188.9	127.1	182.9	106.5	187.9
SD	12.0	12.9	27.5	9.8	11.2	9.7	12.4
CV (%)	9.3	5.9	2.3	7.7	6.1	9.1	6.6
**Within-run** ^b^							
Mean	127.5	214.8	1168.2	127.0	180.3	105.2	178.2
SD	10.8	12.5	15.6	9.1	5.6	6.4	6.8
CV (%)	8.5	5.8	1.3	7.1	3.1	6.1	3.8

CV, coefficient of variation; SD, standard deviation. ^a^ Based on CLSI EP5-A2 tested using three controls, two runs per day in duplicate for 20 days (total *n* = 80). ^b^ Based on one run of 20 tests.

**Table 2 jcm-09-00321-t002:** Distribution of the analytes reported by the ketone body assay in 552 generally healthy, non-fasting adults.

Percentile	Total Ketone Bodies (µM)	β-hydroxybutyrate (µM)	Acetoacetate (µM)	Acetone (µM)
0%	<65.0	<45.0	<26.3	<19.7
2.5%	88.8	48.6	<26.3	<19.7
25.0%	136.9	88.8	<26.3	<19.7
50.0%	174.1	111.3	35.7	26.8
75.0%	235.8	154.1	49.7	37.8
97.5%	623.2	396.4	120.0	78.6
100%	1130	714.7	279.8	271.8

**Table 3 jcm-09-00321-t003:** Baseline clinical and laboratory characteristics in the Insulin Resistance Atherosclerosis Study (IRAS).

Characteristics	Non-Diabetic Subjects (*n* = 825)	T2DM Subjects (*n* = 373)	*p*-Value
Age (years)	55 ± 8	57 ± 8	<0.0001
Sex, men (%)	467 (57)	194 (52)	0.14
Race			
Non-Hispanic white (%)	334 (40)	128 (34)	0.04
Hispanic (%)	272 (33)	119 (32)	0.72
African American (%)	219 (27)	126 (34)	0.01
SBP (mmHg)	122 ± 17	129 ± 18	<0.0001
DBP (mmHg)	78 ± 9	78 ± 9.9	0.42
BMI (kg/m^2^)	28.4 ± 5.6	31.5 ± 5.6	<0.0001
Waist circumference (cm)	90 ± 13	99 ± 12	<0.0001
Fasting glucose (mmol/L)	5.5 ± 0.6	9.7 ± 3.2	<0.0001
2 Hour glucose (mmol/L)	6.9 ± 1.9	17.7 ± 4.8	<0.0001
Fasting FFAs (mmol/L)	0.47 ± 0.19	0.60 ± 0.23	<0.0001
Total cholesterol (mmol/L)	5.44 ± 1.11	5.51 ± 1.12	0.33
HDL-C (mmol/L)	1.21 ± 0.39	1.05 ± 0.30	<0.0001
Triglycerides (mmol/L)	1.24 (0.88–1.81)	1.77 (1.19–2.46)	<0.0001
GlycA (µM)	346 ± 70	373 ± 71	<0.0001
LP-IR score (0–100)	50 ± 21	62 ± 18	<0.0001
Total ketone bodies (µM)	142 (98–208)	182 (116–269)	<0.0001
β-hydroxybutyrate (µM)	100 (67–154)	131 (79–193)	<0.0001
Acetoacetate (µM)	26 (13–44)	32 (15–52)	0.011
Acetone (µM)	15 (8–24)	18 (10–30)	0.0004

Data are the mean ± SD or the median (interquartile ranges). Abbreviations: BMI, body mass index; DBP, diastolic blood pressure; FFAs, free fatty acids; HDL-C, high-density lipoprotein cholesterol; LP-IR, Lipoprotein Insulin Resistance Index; SBP, systolic blood pressure; T2DM, type 2 diabetes mellitus. Triglycerides, insulin, total ketone bodies, β-hydroxybutyrate, acetoacetate and acetone values were log_e_ transformed.

**Table 4 jcm-09-00321-t004:** Univariate analysis of plasma total ketone bodies, β-hydroxybutyrate, acetoacetate and acetone in fasting 1198 subjects (373 subjects with and 825 subjects without type 2 diabetes mellitus) in the Insulin Resistance Atherosclerosis Study (IRAS).

	Total KB		β-HB		AcAc		Acetone	
	r	*p*-Value	r	*p*-Value	r	*p*-Value	r	*p*-Value
Age	0.057	0.050	0.057	0.048	0.028	0.328	0.004	0.884
Sex (men/women)	0.161	<0.0001	0.210	<0.0001	0.014	0.624	−0.028	0.337
Glucose	0.162	<0.0001	0.137	<0.0001	0.139	<0.0001	0.137	<0.0001
FFAs	0.294	<0.0001	0.306	<0.0001	0.155	<0.0001	0.176	<0.0001
TG	−0.078	0.007	−0.099	0.0007	−0.178	0.541	0.047	0.104
LP-IR score	−0.134	<0.0001	−0.157	<0.0001	−0.063	0.028	−0.000	0.990
T2DM (yes/no)	0.152	<0.0001	0.145	<0.0001	0.087	0.003	0.115	<0.0001

Unadjusted Spearman correlation coefficients (r). Abbreviations: AcAc, acetoacetate; β-HB, β-hydroxybutyrate; FFAs, free fatty acids; KB, ketone bodies; LP-IR, Lipoprotein Insulin Resistance Index; T2DM, type 2 diabetes mellitus; TG, triglycerides.

**Table 5 jcm-09-00321-t005:** Multivariable linear regression analysis of plasma total ketone bodies, β-hydroxybutyrate, acetoacetate and acetone in fasting 1198 subjects (373 subjects with and 825 subjects without type 2 diabetes mellitus) in the Insulin Resistance Atherosclerosis Study (IRAS).

	Total KB		β-HB		AcAc		Acetone	
	β	*p*-Value	β	*p*-Value	β	*p*-Value	β	*p*-Value
Age	−0.036	0.188	−0.030	0.273	−0.020	0.494	−0.073	0.011
Sex (men/women)	0.013	0.820	0.064	0.265	−0.079	0.191	−0.136	0.026
FFAs	0.326	<0.0001	0.336	<0.0001	0.215	<0.0001	0.199	<0.0001
TG	−0.132	<0.0001	−0.168	<0.0001	−0.035	0.301	0.019	0.581
LP-IR score	−0.115	0.0005	−0.108	0.0009	−0.110	0.001	−0.072	0.037
T2DM (yes/no)	0.231	0.0003	0.240	0.0001	0.131	0.049	0.168	0.012

β: standardized regression coefficient. Abbreviations: AcAc, acetoacetate; β-HB, β-hydroxybutyrate; FFAs, free fatty acids; KB, ketone bodies; LP-IR, Lipoprotein Insulin Resistance Index; T2DM, type 2 diabetes mellitus; TG, triglycerides. Independent variables included are age, sex, FFAs, TG, LP-IR score and T2DM categorization.

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
