# Peer review of "Ketone Bodies Are Mildly Elevated in Subjects with Type 2 Diabetes Mellitus and Are Inversely Associated with Insulin Resistance as Measured by the Lipoprotein Insulin Resistance Index"

_jcm, 2020, doi:10.3390/jcm9020321_

Round 1
Reviewer 1 Report
The manuscript by Garcia et al. uses an NMR-based quantitative assay for the determination of ketone bodies. At least three items of useful information are presented: the assay itself (which is validated well in the manuscript), reference values for ketone bodies, and a number of relevant associations established using clinical samples. It is worth noting that other NMR-based metabolomics studies have comprised ketone body quantitation as well (eg a recent study by Wang et al. in BMC Med), but the data and associations reported are undoubtedly worth publishing.
Experimental approaches are largely well explained; however, the authors do not explain how exactly their algorithm determines the peak area in the presence of high background signal. This information needs to be added.
Furthermore, I suggest to add information to the introduction that explains the motivation for including FFAs in the analysis (beyond the fact that ketone bodies are derived from a branch of fatty acid metabolism) – which hypotheses are going to be tested here? Also - what is known about links between FFAs and insulin resistance?
It is probably not surprising that levels of ketone bodies correlate positively with high levels of free fatty acids, and – since correlations between high FFA and type 2 diabetes are also well established - with type 2 diabetes. What seems to be puzzling is that levels of ketone bodies are negatively correlated with the LP-IR score – which, unless there is a misunderstanding, might suggest that a high level of ketone bodies, even when going with high FFA and type 2 diabetes, corresponds to low insulin resistance? This seems at the very least counter-intuitive, and therefore needs additional analysis and comment in the discussion. Perhaps this might even turn out the most intriguing part of this study, as it is not necessarily expected. The hint that this finding might advocate a ketogenic diet is insufficient. Is there any way to stratify the patient data to see in which scenarios the ketone bodies-insulin resistance correlations are strong, and whether there are other scenarios where this association does not hold?
In addition, I find the title somewhat confusing, as it seems to suggest that ketone bodies associate in the same manner with FFAs and IR, whilst in fact they behave in the opposite way.
In summary, there is merit in these data, but additional analysis to address the more unexpected association (inverse for ketone bodies/insulin resistance) is recommended.
Minor issues:
Line 193: “15 or more” is ambiguous – can this be specified more clearly?
Table 2: The first column should have a heading. Is this presentation of percentiles common?
Line 339: This is unclear. If this second cohort were patients – how could these “confirm” reference intervals? I would have thought that reference data should be derived from healthy individuals, or at least individuals who do not suffer from diseases that are related to the measured quantity.
Reviewer 2 Report
Overall the manuscript was well written. A thorough validation of using NMR for assessing these ketone bodies at low levels appears missing in the literature and as such is useful. However, both the clinical association results and methods have caused me to question the validity of the results and as such ask the following.
1. It is somewhat confusing that sometimes it is mentioned that ketone bodies are associated with insulin resistance (lines 423, 430) and sometimes inversely associated (lines 39, 411). They are elevated in T2D (Table 3) and yet you find an inverse association with LP-IR score. This appears contradictory. Is your regression analysis a stepwise analysis? Do ketone bodies correlate with LP-IR (correlation analysis instead of multivariate regression)?
2. It also appears unexpected that the generally healthy cohort (Table 2) have a ketone body median value that is between the IRAS non-diabetic and T2D (and in fact closer to the T2D median). Were these processed in exactly the same way to make the calibration valid? Were the IRAS and healthy cohort samples both in same fasted/nonfasting state? If not, then are the healthy cohort valid for use as a reference interval?
3. In Fig 2 the non-spiked sample peaks of acetone AcAc (and B-HB) appear very small if not undetectable to the eye. Although Fig 2 is nice in that it shows the whole spectrum and also the inserted regions of interest (with the spiked sample in comparison to non-spiked), it is also important to see a scale where it is possible to see the signal to noise of the non-spiked AcAc, acetone, B-HB that you are fitting. If the signal to noise is low then all your data would be significantly aided by an increase in number of scans (12 is small).
4. It is unclear exactly how you exactly ‘fitted’ the spectra. The software used should be stated. You mention signal deconvolution analysis – presumably this determines the baseline of backbone residues etc from the peaks of interest. Then how were the areas under the ketone bodies quantified? By an integral between certain frequency ranges or by fitting line shapes (eg. Lorentzian/Gaussian)? In the time or frequency domain? A figure showing (as mentioned above) a non-spiked sample, then it deconvolved and possibly fitted would be good.
5. Although it is good to see both inter- and intra- assay coefficients of variation at different levels, it would be helpful if these (or the low levels) would be of similar concentrations to the data of IRAS shown here in order to be meaningful.
Minor points:
6. What are the approximate T1s (relaxation times) of these metabolites? Would you expect any differences in T1 between serum samples and if so then could this be effected by using a TR of 2.95 s?
7. The positions and multiplicities of acetone, AcAc, and B-HB appear clear from the insert in Fig 2 (or could be clearer if coloured) and so Figure 1 seems redundant.
Round 2
Reviewer 2 Report
No comments (alterations are good and provide clarity; the results are interesting).